# Temperature Impacts on Endurance and Read Disturbs in Charge-Trap 3D NAND Flash Memories

**DOI:** 10.3390/mi12101152

**Published:** 2021-09-25

**Authors:** Fei Chen, Bo Chen, Hongzhe Lin, Yachen Kong, Xin Liu, Xuepeng Zhan, Jiezhi Chen

**Affiliations:** 1School of Information Science and Engineering, Shandong University, Qingdao 266237, China; 202032768@mail.sdu.edu.cn (F.C.); 15763534878@163.com (B.C.); 201932552@mail.sdu.edu.cn (H.L.); kongyachen@126.com (Y.K.); 201700121107@mail.sdu.edu.cn (X.L.); zhanxuepeng@sdu.edu.cn (X.Z.); 2State Key Laboratory of High-End Server & Storage Technology, Testing and Evaluation Research Department, Jinan 250000, China

**Keywords:** 3D NAND flash memory, temperature, endurance, read disturb

## Abstract

Temperature effects should be well considered when designing flash-based memory systems, because they are a fundamental factor that affect both the performance and the reliability of NAND flash memories. In this work, aiming to comprehensively understanding the temperature effects on 3D NAND flash memory, triple-level-cell (TLC) mode charge-trap (CT) 3D NAND flash memory chips were characterized systematically in a wide temperature range (−30~70 °C), by focusing on the raw bit error rate (RBER) degradation during program/erase (P/E) cycling (endurance) and frequent reading (read disturb). It was observed that (1) the program time showed strong dependences on the temperature and P/E cycles, which could be well fitted by the proposed temperature-dependent cycling program time (TCPT) model; (2) RBER could be suppressed at higher temperatures, while its degradation weakly depended on the temperature, indicating that high-temperature operations would not accelerate the memory cells’ degradation; (3) read disturbs were much more serious at low temperatures, while it helped to recover a part of RBER at high temperatures.

## 1. Introduction

After a decade of rapid technological developments, 3D NAND flash memory has been widely utilized in various kinds of storage applications, especially in file memory-related products such as laptops and data centers. Due to the fact of its ultra-high bit density, lower bit cost, and better performances as well as reliabilities, 3D NAND flash is substituting its 2D counterpart step by step. For charge-trap (CT) 3D NAND flash memory, the endurance can largely be improved because the effects of the tunneling layer degradations are weak, and the program time can be faster because the effects of inter-cell interference (ICI) are well suppressed with larger cell-to-cell space. Recently, a 3D NAND with more than 170 layers was announced by the NAND flash makers [1,2]; more impressively, quadruple-level-cell (QLC, 4 bits/cell), penta-level-cell (PLC, 5 bits/cell), and even hexa-level-cell (HLC, 6 bits/cell) operation modes have been demonstrated [3,4]. All these fundamental developments as well as design-technology co-optimizations (DTCO) will drive 3D NAND flash to the mainstream non-volatile memories in the near future [5].

In NAND flash-based memory systems, a major issue that affects both the performance and reliability is the temperature. In conventional 2D NAND flash with a floating gate (FG) structure, as operation temperature increases, the raw bit error rate (RBER) will increase, and the degradation will become more serious. Therefore, on the one hand, the temperature monitor and controller are necessary in NAND flash-based memory systems with robust reliabilities, and on the other hand, the temperature dependence can be utilized as an accelerator to build a lifetime prediction model in a short-time using the Arrhenius model [6], which shows high accuracy in 2D FG NAND flash. However, due to the special structures of cells and the arrays in CT 3D NAND flash, the failure mechanisms are much more complex, and the temperature effects are quite different. This can explain why the conventional lifetime prediction model loses accuracy when it is applied to 3D NAND flashes [7]. Accordingly, comprehensive understandings of the temperature impacts on a 3D NAND flash are strongly required.

In this paper, systematic characterizations of the temperature dependences have been conducted on CT-type 3D TLC NAND flash memories from −30~70 °C, with focus on the RBER modulations in P/E degradations and read disturbs. By using the FPGA-based raw NAND chip tester, it was experimentally observed that the program/erase time and the RBER in the P/E cycling and read disturbs were highly dependent on the temperature. However, temperature had negligible impacts on the cells’ degradation, indicating that the CT 3D NAND is suitable for work at high temperatures with no need to worry about accelerated degradations. 

The main contributions of this paper are as follows:We characterized the P/E cycling in CT 3D NAND flash memory from −30~70 °C using the raw NAND chip tester. An effective TCPT model was proposed to simulate the program time changes by the P/E cycles and the temperature;We characterized the cross-temperature measurements to study the temperature-dependent degradations, indicating that high-temperature operations will not accelerate the degradation of the memory cells;We characterized the measurements of temperature-dependent read disturbs. It showed that read disturb degrades at cold temperature, but it helps to recover a part of RBER at high temperatures. The underlying origins are analyzed in detail.

The rest of the paper is organized as follows: Section 2 introduces the background and related work; Section 3 presents the evaluation setup; Section 4 describes the measured results of P/E cycling; Section 5 shows the temperature-dependent read disturbs; finally, Section 6 concludes this work.

## 2. Background and Related Work

In 2D NAND flash memory, memory cells use FG to store electrons. However, as scaling the cell size to sub-1X nm, serious issues occur from the large cell-to-cell interference and variations in stored charges in FG. It turns out to be extremely difficult to increase the bit density while also guaranteeing the reliability. Different from 2D NAND flash, 3D NAND flash utilizes stacked storage layers to increase the bit density, which settles the problem of flat area scaling and, thus, the key issue turns to be how many layers can be stacked. The comparisons between the two different NAND flash structures are shown in Figure 1a. In a 3D CT NAND flash cell, besides the core oxide and poly-Si channel, the gate stack contains an oxide tunneling layer, silicon–nitride CT layer, oxide blocking layer, and a control gate (CG).

In NAND flash operations, there includes single-level-cell (SLC, 1 bit/cell), multiple-level-cell (MLC, 2 bits/cell), triple-level-cell (TLC, 3 bits/cell), QLC, PLC, and HLC. For 2D NAND flash, SLC and MLC modes are adopted in accordance with the products’ requirements; while for a 3D NAND flash, TLC mode has been widely used because of the well-tuned reliability and high cost–performance ratio. A typical threshold voltage ($V_{th}$) distribution of TLC in a 3D CT NAND flash is shown in Figure 1b, wherein seven program states from A to G levels can be well distinguished. Only a part of the erase state (ER) can be observed due to the negative $V_{th}$ values. Each word-line (WL) consists of three pages, the most significant bit (MSB) page, the central significant bit (CSB) page, and the least significant bit page (LSB). When reading the data from the chip, error bits occur at the overlapping regions between the neighbor states, such as A(B) to B(A) error bits when reading the CSB page at V2 level. Thus, suppressing overlapping regions or optimizing reading voltages are challenges to designing highly reliable NAND flash memory. In addition, for a 3D CT NAND flash, one special concern is the shared common CT layer between neighboring cells. 

This makes the reliability mechanisms more complex because the spatial redistributions of stored charges will seriously affect the data retention and read disturbs [8]. It was also reported that the P/E cycling stress affects the charge redistributions more seriously [9]. In other words, in a 2D FG NAND flash, the failure mechanism is simple, and the temperature effects can be monitored, while for a 3D CT NAND flash, the special cell structure makes the failure mechanisms complex and previous temperature-related models are no longer suitable. In the following, several related works are briefly described.

The 3D NAND flash was developed more than ten years ago in 2007 [10], and the first TLC 3D BiCS flash memory with 32 stacked layers was demonstrated by Toshiba in 2015 [11]. Currently, 174 staking storage layers [1,2] as well as HLC operation mode [4] have been realized and demonstrated. So far, 3D NAND flash has been utilized in many kinds of storage products, especially in smartphones, personal computers, and data centers. To assure the robust reliability and high performance of those products, environmental temperature effects should be well considered, and the effects should be included in the system design. Cai et al. studied MLC NAND flash memory and noticed that the error bits from read disturb were much more likely to take place in cells with lower $V_{th}$ values [12]. Zambelli et al. studied cross-temperature effects in 2D and 3D NAND flash memories and found that there was a large number of fail-bits when the memory was read at a temperature different from that exercised during the program [13]. Wu et al. found that cell $V_{th}$ values had various offset and velocities for different temperate operations, and it can be reduced by shortening the interval time from erase to program during cross-temperature write–read stages [14]. Kong et al. studied the read disturbs in a 3D CT NAND flash memory, and observed that read disturbs were strongly correlated to retention time and temperatures, and proposed the schemes of precharge-the-storage-layer (PCSL) and thermally-stabilize-the-storage-layer (TSSL) to suppress read disturbs [8]. Luo et al. observed that the temperature effects increase retention loss speed at a super-linear rate and increases program variations and concluded that prior models for planer 2D NAND flash were not suitable for 3D NAND flash [15]. Resnati et al. investigated the temperature dependence of cell $V_{th}$, string currents, and random telegraph noise (RTN) distributions in 3D NAND, showing that $V_{th}$ distributions will be tight at high temperatures but widened at low temperatures [16]. In order to minimize the $V_{th}$ distribution widening at low temperature and cross-temperature operations, Venkatesan et al. reviewed the 3D NAND technologies and pointed out that polysilicon channel engineering was necessary [17]. H. Shin et al. investigated the dominant failure mechanisms in 3D NAND after cycling [18] and drew their conclusions that failure mechanisms in 3D NAND are complex; it is not reliable to use temperature as an accelerator for burn-in tests on the basis of the Arrhenius model [7].

As a key factor impacting on NAND flash performance and reliability, temperature effects should be well understood. Thus far, in previous reports, temperature effects have been discussed from material-to-device viewpoints or system-level viewpoints, and most of them focused on data retention. It is necessary to have a comprehensive understanding of the temperature effects to conduct device-to-technology co-optimizations (DTCO) and to provide fundamental information for NAND-based applications in a wide temperature range. In this paper, using a high-performance raw NAND chip tester, temperature-dependent characterizations were performed from −30 to 70 °C by focusing on the RBER in P/E cycling, read disturbs, and cross-temperature degradations.

## 3. Evaluation Setup

Raw NAND chips were characterized using an FPGA-based raw NAND chip tester with eight parallel sockets and high-speed PCIE interfaces with a maximum data transfer speed up to 200 MHz. The tester was specially designed with a capability to withstand wide temperature operations from −40 to 100 °C, and a customized software was used as the interface to carry out data program/erase/read scripts and detailed data analysis. Moreover, a high–low temperature test chamber with precise controllability to the temperature and humidity was used to perform cross-temperature tests. In our experiments, we chose the 3D CT TLC NAND flash memory chip with 64 stacking storage layers, 5912 valid blocks, with block containing 768 logical pages with 18,336 bytes per page [19]. As shown in Algorithm 1, experiment processes were divided into the following:P/E cycling: With combined “Block Program” and “Block Erase” scripts, random data were programmed to the NAND chip and then erased alternately. Here, the generated random data were different in each P/E cycle to ensure the randomness of the characterizations for fair analysis;Read Disturb: After data programming, repeated data reading operations were transferred to the tester controller to perform block data reading. The data were not dumped to the controller until we performed a “Block Read Dump” script;Data Analysis: The programmed data was read out to the customized software using the “Block Read Dump” operation, and data in the TLC NAND chips were downloaded to a text file. By comparing the programmed data and the read-out data, error bit information could be analyzed.


## 4. P/E Cycling

The NAND flash memory tested in this experiment was a 3D CT TLC NAND flash chip, and the stored data were divided into eight states according to the $V_{th}$ of the storage cell. Ideally, the $V_{th}$ between adjacent states has a wide read margin, but the $V_{th}$ of adjacent states had overlapping regions in practice, and these overlapping regions were the source of error bits. It should be noted, due to the limited memory window for programming in a high-bit density NAND flash, such as TLC mode, overlapping regions do exist as shown in Figure 1b. The experimental procedure was as follows: firstly, we raised the temperature to the set value; then, we performed 3000 P/E cycles in randomly selected blocks in the chip, and several blocks were characterized to make sure that our results were reliable and stable; finally, the program/erase times of each P/E cycle was recorded in real-time, and the data were exported for analysis.
**Algorithm 1. The process of experiment.****Definitions:**1: $P_{n}$: the number of P/E cycles, be initialized to 0; 2: T: the temperature of experiment;3: $T_{target}$: the target temperature of experiment; 4: $R_{count}$: the number of read cycles, be initialized to 0; 5: MAX $R_{n}$: the maximum number of read cycles; **Process:**6: **if** T < $T_{target}$ or T > $T_{target}$ **then**7:    Raise T to $T_{target}$;8: **else**9:    Wait 5 min;10:     **for** $P_{count}$ ≤ $P_{n}$ **do**
11:       Execute erase/program operation; 12:       $P_{count}$ + = 1; 13:       Collect program time and erase time;14:       **if** $P_{count}$ == 1 || $P_{count}$ % 200 == 0 **then**15:          Excute the dump operation;16:          Calculate RBER;17:       **end if**18:    **end for**
19:    Change NAND block;20:    **While** $R_{count}$ ≤ MAX $R_{n}$ **do**21:       Excute sequential read operation on NAND block;22:       $R_{count}$ + = 1;23:       **if** $R_{count}$ == 1 || $R_{count}$ % 100 == 0 **then**24:          Excute the dump operation;25:          Compare and calculate RBER;26:          Collect error classification;27:       **end if**28:       **end while**29: **end if**


Figure 2 shows the program time ($t_{prog}$) and the erase time ($t_{erase}$) when performing P/E cycling at various temperatures. In the case of $t_{erase}$, it had a strong dependence on the temperature, but the effects of P/E cycles were negligible. The higher the temperature, the shorter $t_{erase}$. However, $t_{prog}$ depended on both P/E cycles and the temperature as shown in Figure 2b. For blocks after a certain number of P/E cycles, the higher ambient temperature of the NAND flash memory, the less time it took to execute the program operation. While at the same ambient temperature, $t_{prog}$ decreased with P/E cycles. Thereby, we propose a temperature-dependent cycling program time (TCPT) model on the basis of following equation:(1)$$t_{prog}=\alpha \left(T\right)\cdot n_{pe}+\beta \left(T\right)$$
(2)$$\alpha \left(T\right)=k_{1}\cdot T+k_{2},\;\beta \left(T\right)=k_{3}\cdot T+k_{4}$$

*T*, $n_{pe}$, and $t_{prog}$ are the temperature, P/E cycles, and program time, respectively. α(T) and β(T) are the temperature-related functions with fitting parameters $k_{n}$ (*n* = 1~4). As shown in Figure 2b, for blocks with higher than 100 P/E cycles, simulation curves agreed well with the experimental data, indicating that the model was effective at predicting the program latency in a wide temperature range. The values of the fitting parameters are listed in Table 1.

Figure 3 shows the measured RBER when the P/E cycling was performed at different temperatures. No matter what the temperature we choose, RBER has an initially higher value at the fresh state with a decreasing trend in sub-200 P/E cycles, and then it increases during subsequent P/E cycling. Initial higher RBER could possibly come from the unstable initial stage after factory, and the gradually increased RBER can be explained by P/E cycling stress caused cell degradation. During the P/E cycling, defects generated in both the tunneling layer and CT layer and the stability of the NAND cells became worse. The most important thing was that RBER could be suppressed at high temperatures, while it degraded at cold temperatures. These results can be explained by the $V_{th}$ distributions’ changes that depended on the temperature. According to the simulated results by Resnati et al. [16], $V_{th}$ distributions are tight at high temperatures (narrower overlap regions cause fewer error bits) but widened at cold temperatures (wider overlap regions cause more error bits). Furthermore, by normalizing the RBER, it was observed that, although RBER increasing at a rate of 70 °C was greater than that at 0 °C, the degradation tendencies did not show a clear temperature dependence in the whole temperature range as shown in Figure 3b. Considering that different V*_th_* distributions had different sensitivities to the cells’ degradations, it was necessary to use a unified criterion to study the cells’ degradations by cycling stress at different temperatures.

Next, for further evidence, cross-temperature characterizations were conducted to study the temperature effects on P/E cycling stress-related cell degradations. As shown in Figure 4, we designed the following experiment: firstly, we selected three groups of blocks, and all groups performed 1000 P/E cycles at 25 °C (Stage-1); secondly, 1000 P/E cycles were executed in three groups, −30, 25, and 70 °C (Stage-2); then, the temperatures were lowered to 25 °C, and 1000 P/E cycles were subsequently performed on all groups (Stage-3). Finally, each group operated 3000 read cycles at 25 °C. It can be observed that, no matter what the temperature we chose in stage-2, the RBER degradation tendency trends were almost the same in stage-3 and read cycles after cross-temperature P/E cycles. On the one hand, it can be concluded that thermal experiences (up to 70 °C) have negligible impacts on RBER degradation; on the other hand, the 3D NAND is suitable for high-temperature operations because the RBER is lower and the effects of P/E cycling caused damage will not be accelerated at higher temperatures. It should be noted that no matter what the operation mode we adopted, operations at higher temperatures did cause larger degradations to memory cells such as enhanced interface trap generation. Fortunately, these damages do not cause worse error bits degradation using the same criteria (Stage-3) as shown in Figure 4.

## 5. Read Disturb

It is known that high temperature can accelerate the speed of lateral charge migration in the storage layer and modulate the threshold voltage distributions of memory cells [8]. These factors can cause read disturb properties to become more complex at various temperatures. To understand the temperature impacts, we designed the following experiment: firstly, setting the work temperature of the chamber to the target temperature ranging from −30 to 70 °C; then, programming randomized data with subsequent 3000 times read cycling. For detail analysis, data were dumped and recorded every 100 times.

Measured read disturbs are summarized in Figure 5. Firstly, it was observed that RBER degradation turned out to be much more serious at cold temperatures; secondly, for high-temperature operations at 70 °C, a part of RBER can be recovered after serval times reading. Considering that the total RBER included two parts, down-shift errors from the charge loss and up-shift errors from charge accrual, we divided the total error bits to two groups for in-depth analysis: down-shift errors and up-shift errors. As shown in Figure 6, for read disturb-related RBER degradations, down-shift errors were the dominant part with clear temperature dependences, indicating that RBER changes mainly originated from the charge loss. Down-shift error degradation was much stronger at sub −25 °C, but it could be well suppressed at high temperatures, which can be explained by the narrower $V_{th}$ distributions at high temperatures [16]. The interesting phenomenon was that read disturbs from up-shift errors showed the opposite tendency while increasing the operating temperature. For read cycling at 70 °C, up-shift error bits can be partly recovered with read cycles. It was noticed that the decreasing error bits were mainly from cells with high program levels, like F-to-G errors in F-level cells. It should be noted that, as shown in Figure 6f, lower F-to-G error bits can be observed in the whole temperature range from −30 to 70 °C. However, G-to-F up-shift error bits are largely suppressed at 70 °C. Thus, with combined down-shift and up-shift errors, we observed abnormal “recovery” at 70 °C while performing read cycling.

For further understandings, the word-line (WL) dependences of fail bit counts (FBCs) change at 70 °C were studied in detail. By comparing the data from the 1st and 3000th read cycles, it was observed that the dependence of the major state error decreased on the WL index. As shown in Figure 7, error bits from D-E, E-F, and F-G errors showed obvious decreasing trends in higher WL index, and each read cycle in this experiment followed the same observation. In other words, the WLs of the middle-to-low index were the dominant origins for the lower up-shift errors that were attributed to the observed error bits “recovery” at 70 °C.

## 6. Conclusions

In this work, to achieve deep insights into the temperature impacts on the reliability properties of the 3D NAND flash, the TLC (3 bits/cell) 3D CT NAND flash memory chip was tested from −30 to 70 °C using the FPGA-based raw NAND chip tester together with the temperature-controllable chamber. With comprehensive characterizations, firstly, it was observed that program time had a clear dependence on both temperature and P/E cycles by which the TCPT model was proposed; secondly, it was found that RBER can be well suppressed at high temperatures and it degrades obviously at low temperature; then, by the designed cross-temperature measurements, it was found that thermal experience had negligible impacts on RBER degradation; finally, as for read disturbs, it was concluded that read disturbs cause more RBER degradations at cold temperatures while part of RBER can be recovered by read disturbs at high temperatures.

## Figures and Tables

**Figure 1 micromachines-12-01152-f001:**
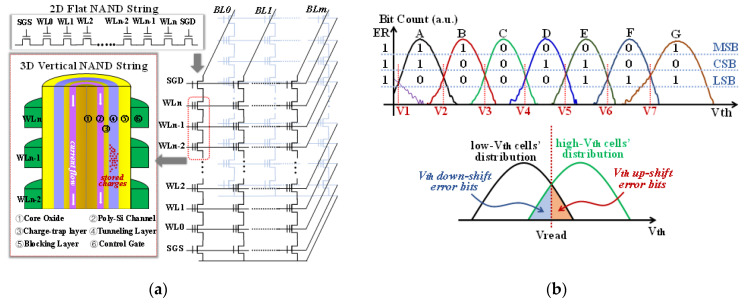
(**a**) A schematic of a 2D NAND string and a 3D NAND string, wherein a 3D NAND cell unit contains core oxide, poly-Si channel, tunneling layer, CT layer, blocking layer, a and control gate; (**b**) TLC operations by storing 3 bits in each cell at three pages: MSB, CSB, and LSB; V1~V7 denote read voltages with the definitions of V*_th_* down-shift and up-shift errors.

**Figure 2 micromachines-12-01152-f002:**
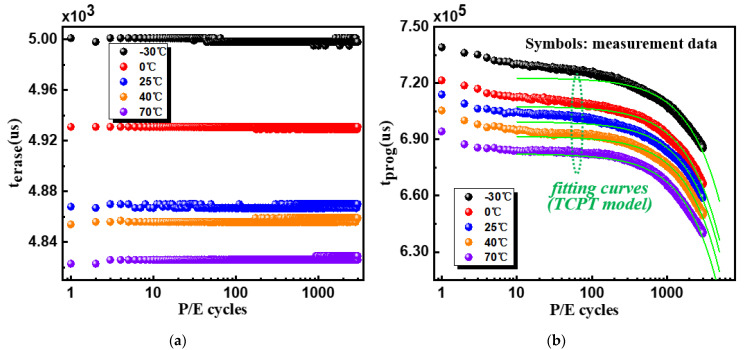
Measured operation times during P/E cycling at different temperatures: (**a**) erase time ($t_{erase}$) and (**b**) program time ($t_{prog}$). $t_{erase}$ depends on the temperature, while $t_{prog}$ depends on both the temperature and P/E cycles, which agreed well with the simulation curves in the blocks with higher than 100 P/E cycles.

**Figure 3 micromachines-12-01152-f003:**
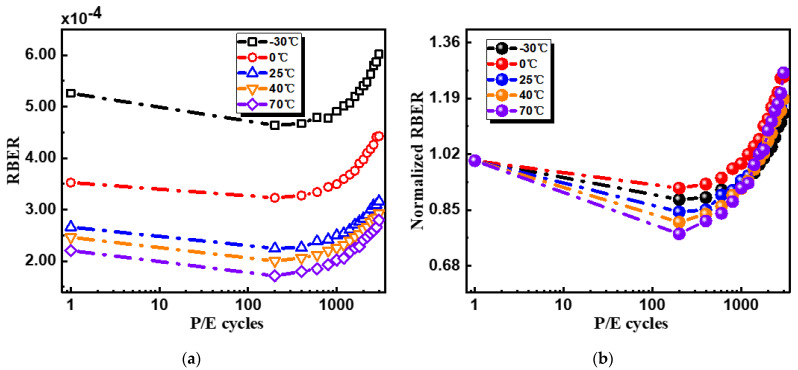
(**a**) Measured RBER with P/E cycling at different temperatures; (**b**) normalized RBER to study RBER degradation.

**Figure 4 micromachines-12-01152-f004:**
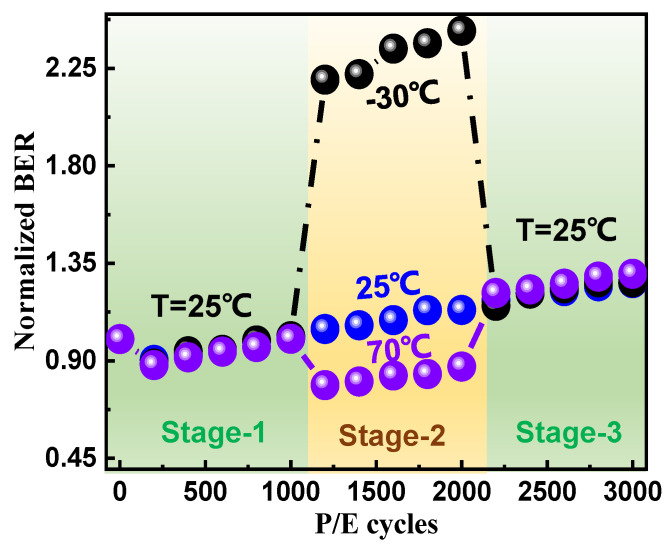
Cross-temperature characterizations to study degradation at various temperatures. The first and third stages were fixed to 25 °C, while the second stage selected three different temperatures, −30, 25, and 70 °C. The RBER was normalized by the first point of the RBER to compare the degradation trends of each condition.

**Figure 5 micromachines-12-01152-f005:**
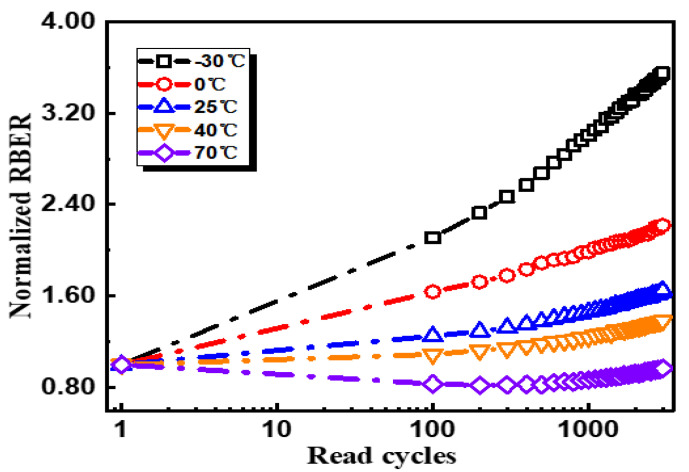
Read disturb characterizations at different temperatures from −30 to 70 °C.

**Figure 6 micromachines-12-01152-f006:**
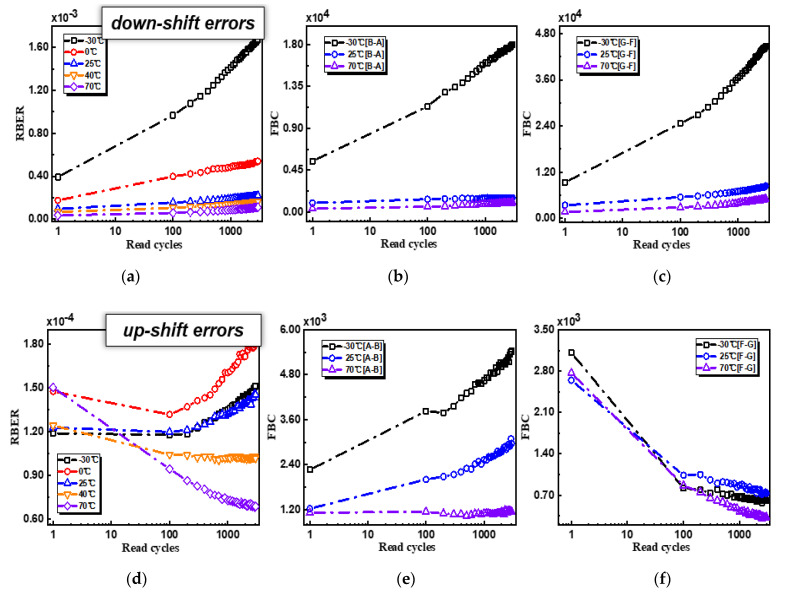
Read disturb-related RBER changes were divided into (**a**) down-shift errors and (**d**) up-shift errors from −30 to 70 °C; (**b**,**c**) compares B-to-A errors and G-to-F down-shift errors, respectively, while (**e**,**f**) compares A-to-B errors and F-to-G up-shift errors, respectively, at −30, 25, and 70 °C.

**Figure 7 micromachines-12-01152-f007:**
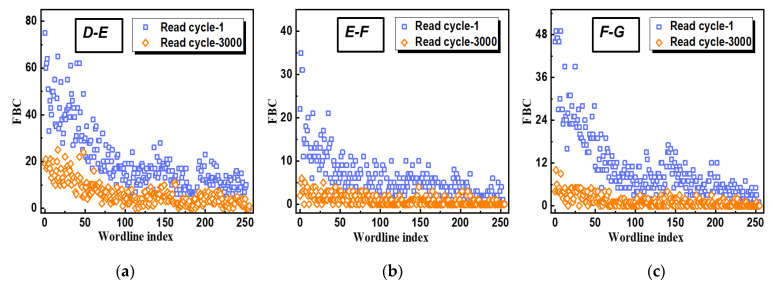
Measured fail bit count (FBC) of different program levels: error bits from (**a**) D-to-E; (**b**) E-to-F; (**c**) F-to-G.

**Table 1 micromachines-12-01152-t001:** The values of fitting parameters in α(T) and β(T).

Fitting Parameter	$k_{1}$	$k_{2}$	$k_{3}$	$k_{4}$
α(T)	−0.0214	−0.13205	/	/
β(T)	/	/	−391.98	707913

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
