# Peer review of "Temperature Impacts on Endurance and Read Disturbs in Charge-Trap 3D NAND Flash Memories"

_micromachines, 2021, doi:10.3390/mi12101152_

Round 1
Reviewer 1 Report
Summary
This paper performs actual experiments on a 3D NAND flash chip to understand how the temperature will affect the program time, erase time, RBER, and read disturbance.
Comments
- For today's NAND flash chip, methods have been proposed to further extend their lifetime via changing from TLC mode to MLC/SLC mode, could the author comment on how the temperature will affect NAND flash if it's in MLC/SLC mode?
- There is a formulation given by the authors on page 6. The fitting parameters should be given to make the results reproducible.
- Could the author comment on how the temperature will affect the wearing of NAND flash? Does it cause more wearing to a cell when the temperature is high?
- Page 3 lines 126 and 127 should be the same line.
Reviewer 2 Report
1) What is the main motivation of this paper? In introduction, you mentioned several issues of temperature effects in 3D NAND (retention loss, RTN, poly-Si channel, failure mechanism). But these are not discussed in your paper and I don’t understand why P/E times and RBER with respect to the temperature and P/E, R cycles are analyzed here.
2) Fig. 2b has just TCPT model, not the simulation result. But the TCPT model does not fit the curvature and the flat plateau at the P/E cycles from 10 toward 3000. Can you say that TCPT model fits well? And can you discuss the meaning and significance of fitting parameters as a function of temperature?
3) P/E cycles in Fig. 2 are in steps of 1. Why is this different from Algorithm1? Does the number of P/E cycles affect program, erase times, and RBER with respect to temperature?
4) It is not understood physically that P/E cycling stress affects the 3D NAND equally irrespective of temperature. As the P/E cycling stress induces interface traps and increases the trap-assisted tunneling (TAT), since the TAT is proportional to temperature, 3D NAND would have different outcomes as a function of temperature. What do you think about this?
5) I am not sure the RBER is normalized at the P/E cycle of 1 in Fig. 3b. You mentioned in the script that “initial higher RBER could possibly come from the unstable initial stage after factory.” Then, why did you normalize RBER at that point?
6) Can you explain physically why Vth distributions at high temperature is narrower than those at low temperature? (In my opinion, cold temperature decreases phonon scattering and TAT, so temperature-related Vth variations can be clear, isn’t it?) If the sentence is valid and true, is it better to operate 3D NAND at high temperature? Please explain this.
7) In the line 216 of page 6, “the temperature effects on P/E stress related RBER degradations could be ignorable” seems different from the result in Fig. 3b. Comparing 0 and 70 C, RBER increasing rate at 70 C is greater than that at 0 C. Please explain this.
Round 2
Reviewer 2 Report
NO comments